# Diversification of mandarin citrus by hybrid speciation and apomixis

Guohong Albert Wu [1,8✉], Chikatoshi Sugimoto[2,8], Hideyasu Kinjo[3], Chika Azama [2], Fumimasa Mitsube[3], Manuel Talon[4], Frederick G. Gmitter Jr. [5✉] & Daniel S. Rokhsar [1,2,6,7✉]

The origin and dispersal of cultivated and wild mandarin and related citrus are poorly understood. Here, comparative genome analysis of 69 new east Asian genomes and other mainland Asian citrus reveals a previously unrecognized wild sexual species native to the Ryukyu Islands: *C. ryukyuensis* sp. nov. The taxonomic complexity of east Asian mandarins then collapses to a satisfying simplicity, accounting for tachibana, shiikuwasha, and other traditional Ryukyuan mandarin types as homoploid hybrid species formed by combining *C. ryukyuensis* with various mainland mandarins. These hybrid species reproduce clonally by apomictic seed, a trait shared with oranges, grapefruits, lemons and many cultivated mandarins. We trace the origin of apomixis alleles in citrus to mangshanyeju wild mandarins, which played a central role in citrus domestication via adaptive wild introgression. Our results provide a coherent biogeographic framework for understanding the diversity and domestication of mandarin-type citrus through speciation, admixture, and rapid diffusion of apomictic reproduction.

[1] DOE Joint Genome Institute, Lawrence Berkeley National Laboratory, Berkeley, CA, USA. [2] Okinawa Institute of Science and Technology Graduate University, Onna, Okinawa, Japan. [3] Okinawa Prefectural Agricultural Research Center, Nago Branch, Nago, Okinawa, Japan. [4] Centro de Genómica, Instituto Valenciano de Investigaciones Agrarias, Valencia, Spain. [5] Citrus Research and Education Center, Institute of Food and Agricultural Sciences, University of Florida, Lake Alfred, FL, USA. [6] Department of Molecular and Cell Biology, University of California, Berkeley, CA, USA. [7] Chan-Zuckerberg BioHub, San Francisco, CA, USA. [8] These authors contributed equally: Guohong Albert Wu, Chikatoshi Sugimoto. ✉email: gwu@lbl.gov; fgmitter@ufl.edu; dsrokhsar@gmail.com

Mandarin-type citrus comprise a heterogeneous group of east Asian citrus with small and typically easily peelable fruit[1,2]. Because of their consumer-friendly attributes, mandarins have seen the greatest percentage increase in global citrus production[3].The phenotypic and genomic diversity of mandarin types in the Nanling region of southern China has driven speculations that mandarins first arose and were domesticated in this region[4]. Yet the nature of the domestication process, and the relationships among these mainland Asian types remain poorly understood.

These mysteries are compounded by the extensive mandarin diversity of the Ryukyu islands and mainland Japan[5], including: tachibana [*C. tachibana* (Makino) Yu Tanaka], a culturally significant ornamental citrus grown throughout mainland Japan; shiikuwasha [*C. depressa* Hayata], grown in the Ryukyus and renowned for its health promoting qualities[6]; and other traditional and wild Ryukyuan citrus of unknown ancestry (Supplementary Note 1). Since tachibana and shiikuwasha have been found in wild forests, and are described in the earliest poetry of the Japanese and Ryukyuan kingdoms[7–10], they have been presumed to be indigenous or natural species[1,11], although some researchers have suggested that they are interspecific hybrids of various kinds[1,12–14]. The complexity of relationships among indigenous and cultivated mandarins across east Asia remains unclear, and is a barrier to understanding the origin and domestication of mandarins.

Most domesticated mandarins can reproduce true to type from seed (apomixis) by generating maternal clones from somatic tissue through the process of nucellar embryony[15], which allows desirable genotypes to be replicated at scale. Although apomixis has been shown to be inherited in a dominant Mendelian fashion[15–18], its natural origin and dispersal across diverse citrus are obscure. Mandarins are also widely hybridized with other citrus species to produce a diversity of familiar cultivated varieties including oranges, grapefruit, and lemons[19,20], which also reproduce apomictically.

In order to resolve the relationships among wild and cultivated mandarins and explore the nature, evolution, and biogeography of east Asian citrus, here we present the genome sequences of 69 traditional, wild, and atypical citrus of the Ryukyus and southern mainland Japan (Supplementary Data 1 and 2; Supplementary Note 2), and analyze these genomes together with previously sequenced wild and domesticated Chinese mandarins, including those from Mangshan in the Nanling mountain range, and other citrus[4,20,21] (Fig. 1). We find that the complexity of mandarin relationships is considerably simplified by the discovery of three ancestral lineages which, together with pummelo, gave rise to all extant mandarin diversity by hybridization and introgression. One of these groups is a previously unknown wild species currently found in the Ryukyu islands; the other two are previously unrecognized sister subspecies of mainland Asian mandarin. Our analysis leads to a comprehensive revision of the origin and diversification of east Asian citrus, including the elucidation of the origins of apomixis in mandarin and its spread to related citrus including oranges, grapefruits and lemons.

## Results

***Citrus ryukyuensis* is a new species of mandarin citrus.** Most strikingly, we identified a new wild citrus species native to the Ryukyu islands that we designate *C. ryukyuensis* sp. nov. (Supplementary Fig. 1, Supplementary Note 3). This new species is represented in our collection by eight wild Okinawan accessions that form a cluster of genomes separated from all previously sequenced species of *Citrus* (Fig. 1a). These accessions include 'tanibuta' types ("big seeds" in Okinawan dialect; Supplementary Note 1) that were described by Tanaka[5] as a Ryukyuan variant of

tachibana. We find that *C. ryukyuensis* is related to but genetically distinct from tachibana and shiikuwasha. Among their differences, *C. ryukyuensis* is a sexual species that produces monoembryonic seeds, while tachibana[22] and shiikuwasha[11] produce polyembryonic (apomictic) seeds.

The identification of *C. ryukyuensis* as a pure species (i.e., a distinct sexually reproducing population without admixture) is supported by three findings (Fig. 1). First, this population has low genome-wide heterozygosity (0.2–0.3%) that is comparable to or less than the variation seen within other recognized citrus species[20], and smaller than the typical interspecific variation in citrus[21] (Fig. 1c). Second, *C. ryukyuensis* nuclear and cpDNA types form distinct clades to the exclusion of other mainland Asian citrus species (Supplementary Fig. 2). *C. ryukyuensis* is strongly differentiated from both *C. reticulata* (i.e., mainland Asian mandarin; $F_{ST} = 0.67$) and *C. maxima* (pummelo; $F_{ST} = 0.82$). Based on sequence comparisons, we estimate that *C. ryukyuensis* diverged from mainland Asian mandarins around ~2.2–2.8 Mya (Fig. 2, Supplementary Notes 3, 11). This divergence time is comparable to the split between other recognized citrus species, e.g., *Poncirus trifoliata* and *P. polyandra*[23]. Finally, the allele frequency spectrum in the *C. ryukyuensis* population decays roughly as expected for a panmictic sexual population (Supplementary Fig. 3), an observation that is consistent with monoembryony (Supplementary Fig. 1b) and zygotic (sexual) reproduction.

**Common mandarin and mangshanyeju are two subspecies of mainland Asian mandarin.** We uncovered further surprises when we analyzed the mainland Chinese wild mandarins sequenced by Wang et al.,[4] in our expanded context (Supplementary Note 9). We find that wild mainland Asian mandarins comprise two sister populations with substantial genetic differentiation (Fig. 1, Supplementary Figs. 2 and 4), in contrast to Wang et al.'s description of a single wild population of Chinese mandarins from which domesticated varieties were selected. For taxonomic simplicity, we consider the two sister populations as sub-species of *C. reticulata* (Blanco). One sub-species, which we call 'common mandarin,' is the predominant contributor to domesticated mandarin. Many domesticated types, however, also contain admixture from the newly recognized second mandarin sub-species and from pummelo (Fig. 1b).

We identify the second mainland mandarin subspecies with types that are colloquially referred to as "mangshanyeju", i.e., wild mandarin ("ju") from the Mangshan region of the Nanling mountain range. Although Wang et al.[4] regard wild mandarins as an undifferentiated group, we find that mangshanyeju (MS) and common mandarin (MA) populations are sufficiently differentiated from each other ($F_{ST}$~0.49) that they should be considered as at least distinct sub-species (Supplementary Note 9). We estimate that these two populations diverged 1.4–1.7 million years ago (Fig. 2 and Supplementary Note 11). We find that the collection of Wang et al.[4] includes two pure mangshanyeju (MS1 and MS2) and two distinct F1 hybrids of mangshanyeju with nearly pure common mandarins (M01 and M04) (see Figs. 1 and 4c, Supplementary Fig. 4). Other citrus also have mangshanyeju ancestry (Fig. 1). For example, we find that yuzu, cultivated for its pleasing flavor and aroma valued in gastronomy and aromatherapy, is an F1 hybrid of mangshanyeju with an Ichang papeda seed parent (Supplementary Note 8). We caution that "mangshanyeju" should not be confused with 'mangshanyegan' (wild citrus ('gan') from Mangshan, *C. mangshanensis*), which is a distantly related citrus species from the same region[4,20,21].

The estimated divergence times between *C. ryukyuensis* and *C. reticulata* (2.2–2.8 Mya), and between mangshanyeju and

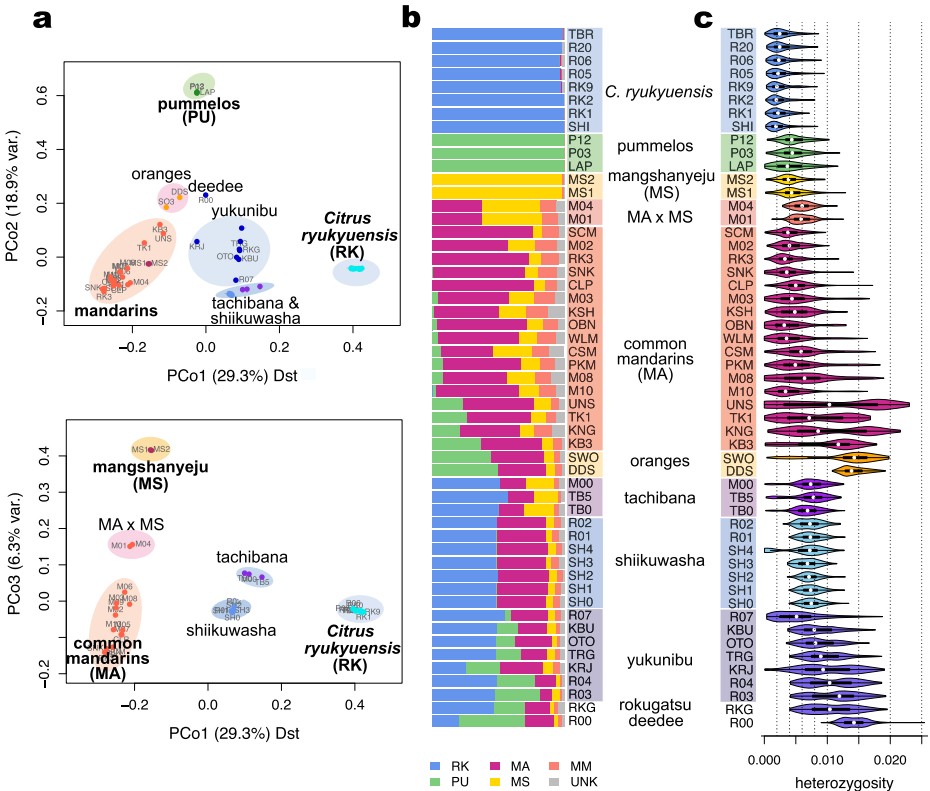

**Fig. 1 Population structure, genetic admixture, and heterozygosity of east Asian citrus. a** Multidimensional scaling (MDS) plot of 51 citrus accessions. Projection onto the first two principal coordinates (upper panel) shows *C. ryukyuensis* as a distinct population from tachibana, shiikuwasha, and other Ryukyuan hybrids (yukunibu and deedee). The third principal coordinate (lower panel) separates the two Mangshan wild mandarins (MS) from other mandarins. It also separates tachibana from shiikuwasha. For easier visualization, accessions with significant pummelo ancestry (pummelos, oranges, some mandarins, yukunibus) are not shown in the lower panel. See Supplementary Data 1 and 3 for accession code and names. **b** Four-way admixture plot of 53 citrus accessions based on local ancestry inference. PU=pummelo (*C. maxima*), RK=*C. ryukyuensis*, MS=mangshanyeju, MA=common mandarin, MM=generic *C. reticulata* without subspecies assignment (MS vs MA), UNK=unknown. Note that tachibana has more MS alleles than shiikuwasha and other Ryukyuan hybrids. Some wild mandarins (M01, M04) are hybrids with nearly equal contribution from the two subspecies of MS and MA. Common mandarins display varying degree of MS admixture. **c** Heterozygosity distribution violin plot for the same accessions as in **b**), for non-overlapping windows of 500,000 callable sites. *C. ryukyuensis* shows the lowest heterozygosity compared to tachibana, shiikuwasha and other hybrid types as well as accessions from *C. reticulata* and *C. maxima*. Median and quartiles are denoted by the white dot and black bar limits respectively, and whiskers are 1.5× inter-quartile range. Source data are provided as a Source Data file.

common mandarins (1.4–1.7 Mya) are consistent with the paleogeology of the region (Fig. 2, Supplementary Fig. 5, Supplementary Note 11). During the early diversification of citrus throughout southeast Asia in the Late Miocene (11.61–5.33 Mya)[20], the boundary of mainland Asia included what is now the Ryukyu arc and the main islands of Japan[24,25]. Sea level rise and tectonic activity isolated the Ryukyu archipelago in the Pliocene (5.33–2.58 Mya) and Pleistocene, with intervals of connectivity to the south near present day Taiwan and north to the Japanese islands. This variable connectivity and associated climatic variation led to the emergence of new species in this region in many plant[26–30] and animal[31–34] taxa, coinciding with our estimates for the divergence of the distinct *C. ryukyuensis* from mainland Asian mandarin. The emergence of *C. ryukyuensis* by allopatric speciation was accompanied by a population bottleneck, suggested by its reduced heterozygosity relative to mainland mandarins (Supplementary Note 11).

**New species illuminates origins of shiikuwasha, tachibana, and other Ryukyu types.** *C. ryukyuensis* provides the key to unlocking the origin and diversity of Ryukyuan and mainland Japanese citrus (Fig. 3). The shiikuwasha in our collection form a large half sibling family with one mainland mandarin parent but distinct

*C. ryukyunensis* parents (Fig. 3, Supplementary Note 4). Unexpectedly, we found a clonal relative of this mainland mandarin parent of all shiikuwasha in a private collection in Nago City, Okinawa (RK3 in our designation, Fig. 3a). RK3 is referred to colloquially as an "ishikunibu" type, but is distinct from the shiikuwasha variety of the same name (Supplementary Note 4). Sequence comparison shows that RK3 is a close relative of the pure[20] Chinese mandarin Sun Chu Sha Kat (SCM) (coefficient of relatedness 0.41), but contains a single 2.4 Mbp introgressed pummelo segment.

The observation that shiikuwasha form a large half-sib family explains the previously puzzling finding that shiikuwasha chloroplast DNAs (cpDNAs) are of two distinct types[35,36], either matching tachibana mandarins (here recognized as *C. ryukyuensis* type), or matching certain mainland Asian mandarins (here recognized as a *C. reticulata* type). Evidently, RK3 mandarin served as both seed and pollen parent in the numerous hybridization events that generated shiikuwasha. The hybrid nature of shiikuwasha accounts for its previously noted genotypic and phenotypic diversity[8,12], and is consistent with previous suggestions that shiikuwashas are hybrids based on high levels of nucleotide polymorphism[1,12,37]. More detailed understanding was elusive since *C. ryukyuensis* had not been recognized or characterized.

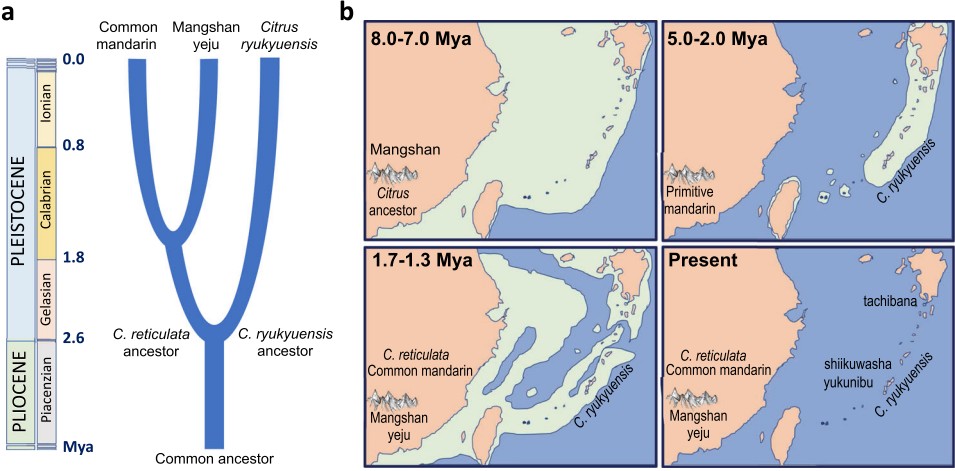

**Fig. 2 Chronogram of east Asian mandarin citrus speciation and biogeography in the Ryukyu Arc and mainland Japan. a** Population divergence times of *C. ryukyuensis* (2.2–2.8 Mya) and two subspecies of mainland Asian mandarins (*C. reticulata*): common mandarin and mangshanyeju (1.4–1.7 Mya). Extant common mandarins are recent admixtures with both mangshanyeju and pummelos. **b** Geological history of the Ryukyu Arc and evolutionary origins of east Asian citrus during four representative time periods: (1) initial radiation of citrus during the late Miocene[20] with subsequent dispersal to regions including Mangshan of the Nanling mountain range. The exact arrival time of primitive mandarins at Mangshan cannot be determined and could be as late as the Pliocene epoch (5.3–2.6 Mya) (top left), (2) geographical isolation and genetic divergence of *C. ryukyueneis* in the Ryukyu Arc from mainland Asian mandarins during early Pleistocene (top right), (3) divergence of mangshanyeju and common mandarins (bottom left), and (4) current distribution of east Asian citrus with *C. ryukyuensis* ancestry in the Ryukyu Arc and mainland Japan, as a result of distinct hybridization events with different migrant mainland mandarins (bottom right). (Maps are adapted from Kimura[25] with paleo-landmasses in light green.) Source data underlying Fig. 2a are provided as a Source Data file.

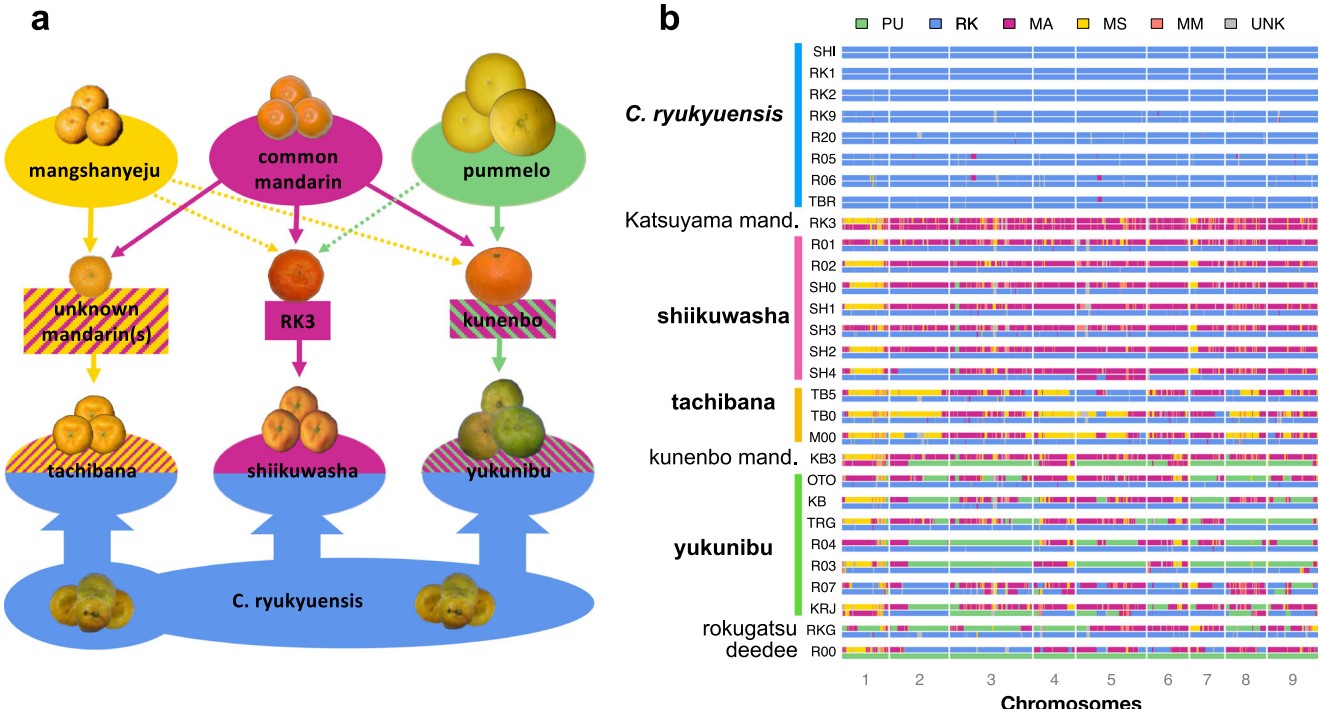

**Fig. 3 Hybrid speciation and admixture map of Ryukyuan and mainland Japanese citrus. a** Origin of Ryukyuan and mainland Japanese citrus types (tachibana, shiikuwasha, yukunibu) derived from four ancestral populations. Thick arrows denote ancestry involving multiple individuals from a population, whereas a thin arrow stands for single individual ancestry. Dotted and solid lines from the top row denote small and significant introgression, respectively. For example, RK3 has small amount of pummelo admixture whereas kunenbo has significant pummelo introgression. The shiikuwashas are half-sibs sharing the same mainland Asian mandarin parent (RK3) but different *C. ryukyuensis* parents. Kunenbo (KB3) is the seed parent of the yukunibu group. **b** Four-way admixture map for Ryukyuan and mainland Japanese citrus types. Population code as in Fig. 1b. Tachibana genomes are characterized by both significant admixture with MS and segments of diploid *C. ryukyuensis* genotype. SH4 is a seedless shiikuwasha. Source data underlying Fig. 3b are provided as a Source Data file.

We find that tachibana is also a collection of hybrids between *C. ryukyuensis* and mainland Asian mandarins, but distinct from shiikuwasha (Fig. 3b, Supplementary Note 5). The extensive sharing of mainland mandarin haplotypes among our tachibana genotypes is consistent with a single Chinese mandarin-like parent although we cannot rule out a small number of closely related mandarin parents (Supplementary Fig. 6). Importantly, the mainland parents of shiikuwasha and tachibana are not related, implying that these geographically separated hybrid species arose independently. In contrast to the simple inter-specific F1 hybrid origin of shiikuwasha, tachibana genotypes are more complex. Each tachibana carries 4-6 multi-megabase segments of diploid *C. ryukyuensis* within an otherwise *C. ryukyuensis* × *C. reticulata* hybrid background, which implies that the direct mandarin-like parents of tachibana themselves had prior introgression of *C. ryukyuensis* (Fig. 3b). Our study shows that tachibana are not generally full siblings, as suggested by an earlier marker-based analysis of three accessions[13].

Finally, much of the remaining diversity of indigenous Ryukuan citrus (including several other named species[5,38–40]) can be organized into a third hybrid family that we named yukunibu, meaning "sour citrus" in Okinawan dialect. Yukunibu citrus are F1 hybrids with a kunenbo-mikan seed parent and diverse *C. ryukyuensis* pollen parents (Fig. 3, Supplementary Note 6). The yukunibu group unites three cultivated accessions (oto, kabuchii, and tarogayo, grown for juice) with two others. While kabuchii's kunenbo-mikan ancestry was previously suggested[13,39], its other familial relationships were not previously recognized. The yukunibu family presumably arose soon after kunenbo-mikan was introduced to the Ryukyus (and then mainland Japan) from Indochina sometime between the 8th and 12th centuries[41–45]. Our collection also contains other members of the extended yukunibu family as well as other unrelated hybrid genotypes with *C. ryukyuensis* ancestry (Supplementary Note 7, Supplementary Fig. 7).

**Apomixis.** While *C. ryukyuensis* is a sexually reproducing species, its hybrid derivatives (shiikuwasha, tachibana, and yukunibu) reproduce apomictically by nucellar embryony. This implies that the apomixis trait was transmitted to these three hybrid species by their migrant mainland mandarin parents. Apomictic reproduction of shiikuwasha[46] and tachibana[22] enabled the rapid establishment and dispersal of these new hybrid species after their formation by hybridization with the pre-existing locally adapted *C. ryukyuensis* population. Notably, the mainland mandarin parents of shiikuwasha and yukunibu (RK3 and kunenbo-mikan) both produce polyembryonic seed[16].

All apomictically reproducing citrus in our collection carry a recently described MITE (miniature inverted-repeat transposable element) DNA transposon insertion in the promoter of the *CitRKD1*[18] gene (also known as *CitRWP*[17]) that dominantly confers an apomictic phenotype. We find that this MITE insertion arose in the mangshanyeju population in the early Pleistocene and spread into other mandarins, oranges, grapefruits, and lemons as well as shiikuwasha, tachibana, and yukunibu (Fig. 4; Supplementary Note 10). The four mangshanyeju-derived MITE haplotypes of *CitRKD1* (from two related groups) occur in different combinations (along with the ancestral allele without MITE insertion) in diverse apomictic citrus, highlighting the complex history of this critical genomic region (Supplementary Fig. 8; Supplementary Data 1 and 3).

Selection for apomixis explains widespread mangshanyeju admixture across cultivated and wild mandarins, especially on chromosome 1 around the *CitRKD1* gene (Figs. 3b, 4c). The adaptive wild introgression of apomixis alleles was a driver of domestication because it allowed the mass production of desirable types. This is consistent with the limited diversity of chromosome 1 haplotypes around the apomixis locus in cultivated mandarin and mandarin hybrids (Fig. 4b). In contrast, the other eight chromosomes in mandarins are typically dominated by alleles from the "common" mandarin sub-population, including those believed to confer low acidity[20,47,48], anthocyanin production[49–51], self-compatibility[43,52], and to regulate flesh and peel color[53–55] and volatile aromatics that contribute to flavor[56].

**Formation of east-Asian mandarin types.** The three native Ryukyuan and Japanese hybrid species with mixed ancestry—shiikuwasha, tachibana, and yukunibu—arose by independent hybridizations of one or a few mainland mandarin migrants with the native wild sexual *C. ryukyuensis*. Since shiikuwasha and tachibana are referenced in native poetry and songs, and were found in wild forests, we suggest that the mainland Asian mandarin founders of these species were chance prehistoric introductions during intervals of connectivity with mainland Asia (40,000–200,000 years ago)[25]. Alternatively, their seeds could have arrived via the Kuroshio Current[34,57], a powerful south-to-north warm current that passes by Taiwan, the Ryukyu Arc and mainland Japan. In contrast, kunenbo-mikan (the mainland Asian parent of yukunibu) was introduced to the Ryukyus by trade between the 8th and 12th centuries[42]. Other imported mainland Asian citrus such as sour orange (daidai) and pummelo also hybridized with *C. ryukyuensis* and shiikuwasha to produce rokugatsu and Ryukyu sour orange (deedee), respectively (Supplementary Fig. 7, Supplementary Note 7).

Although some authors have suggested a close relationship between shiikuwasha and tachibana[1,36], we find them to be distinct hybrid species. Shiikuwasha and tachibana differ both in their mainland Asian mandarin parents and by the differentiation of their *C. ryukyuensis* ancestors. While the mainland Asian mandarin parent of shiikuwasha had prior pummelo admixture but limited mangshanyeju introgression, the unknown mainland Asian mandarin parent(s) of tachibana had substantial mangshanyeju admixture (48–54%, comparable to an F1 hybrid of mangshanyeju and common mandarin) and no pummelo admixture. On the *C. ryukyuensis* side, genetic differentiation among tachibana and shiikuwasha haplotypes ($F$st =0.17–0.20) is consistent with separate mainland Japanese and Ryukyuan *C. ryuykyuensis* populations that diverged ~220,000–350,000 years ago (Supplementary Note 11). The *C. ryukyuensis* haplotypes of shiikuwasha and yukunibu are more closely related to the extant *C. ryukyuensis* population in Okinawa.

## Discussion

Here we have shown that extant mandarin diversity arose from three ancestral populations distinguished by comparative genome analyses: the island species *C. ryukyuensis*, and common mandarin and mangshanyeju from mainland Asia. Complex patterns of admixture involving these previously unrecognized founding populations, combined with clonal propagation by apomixis, produced the extensive heterogeneity of mandarin citrus. Our approach consists of identifying natural or pure species based on distinctive patterns of genetic variation, followed by characterization of hybrid genotypes in terms of these founding types[20,21,58]. These new insights into wild mandarin diversity, coupled with the recognition that widespread pummelo introgression has also contributed to mandarin domestication, enables a comprehensive admixture-informed classification scheme for mandarin citrus[20,59] (Supplementary Note 12). Consideration of domestication phenotypes including nucellar embryony, fruit size and palatibility[20] supports its practical use.

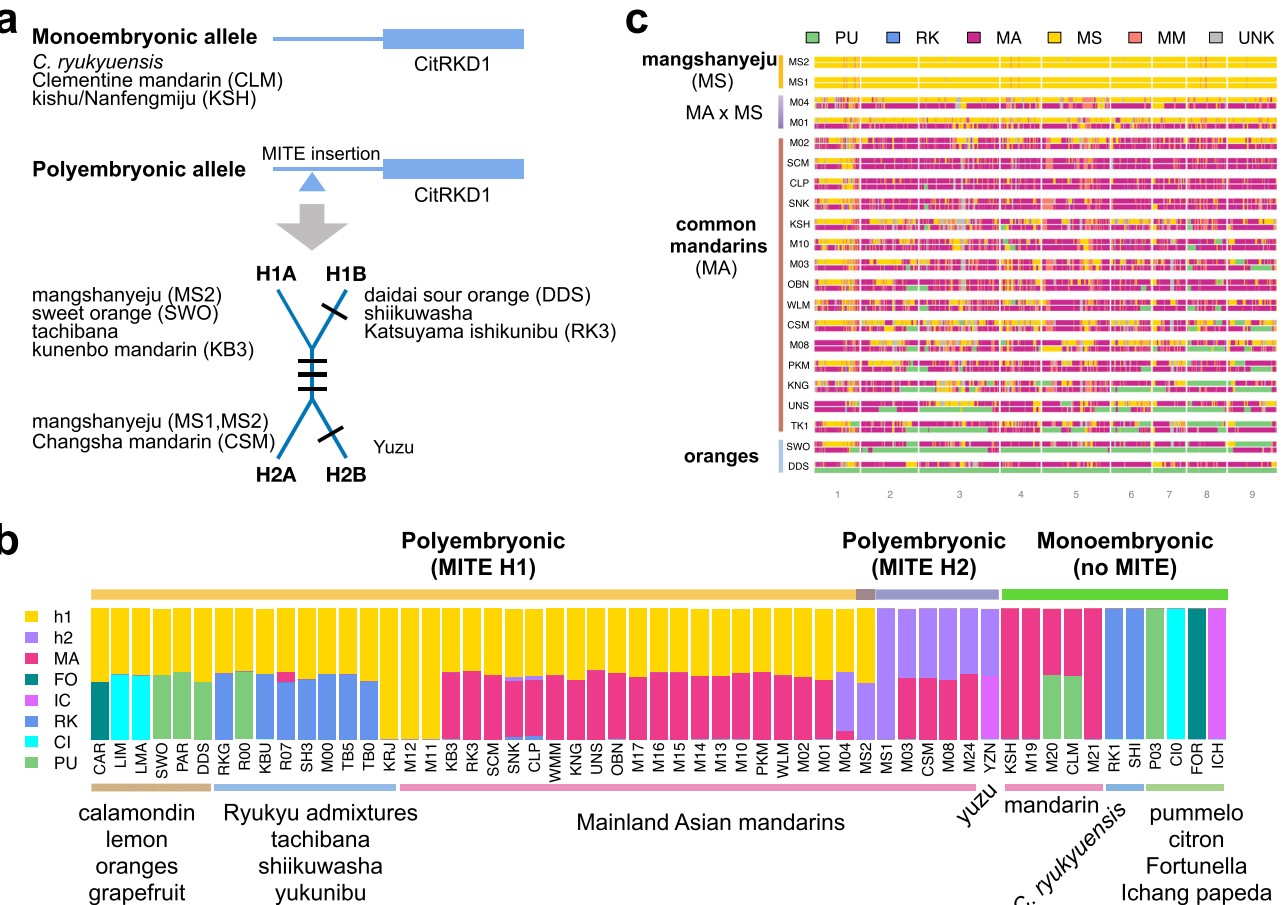

**Fig. 4 Ancestry of apomixis alleles and two subspecies of mainland Asian mandarins (*C. reticulata*). a** Diversity of the apomixis alleles in mandarins and inter-specific mandarin hybrids. The ancestral allele does not have the MITE transposon insertion in the promoter of the *CitRKD1* gene regulating citrus apomixis. Derived alleles with the MITE insertion are dominant for the nucellar embryony phenotype. Four MITE haplotypes in two haplogroups (H1=H1A and H1B; H2=H2A and H2B) are observed among sequenced mandarins and hybrids with each black line denoting a segregating SNP. Listed next to each MITE allele type are representative citrus accessions containing that allele. **b** Genetic ancestry of the citrus polyembryonic locus (200 kb region flanking *CitRKD1* gene). Fifty-five accessions derived from six progenitor species are analyzed with ADMIXTURE[68] and the eight-population (*K*=8) structure is presented with additional figures shown in Supplementary Fig. 8. (PU=pummelo, CI=citron, RK=*C. ryukyuensis*, IC=Ichang papeda, FO=*Fortunella* (kumquat), MA= common mandarin, h1 and h2 have mangshanyeju ancestry). Accessions with h1 ancestry contain MITE H1A or H1B, whereas those with h2 ancestry have MITE H2A or H2B. MS1 and MS2 are two mangshanyeju accessions. All sequenced polyembryonic accessions carry the dominant allele with the MITE insertion and have mangshanyeju ancestry at this locus, whereas monoembryonic accessions have common mandarin but not mangshanyeju ancestry. **c** Genome-wide local ancestry inference of mainland East Asian citrus with four ancestral populations including two subspecies of *C. reticulata* (MS, MA). Population code as in Fig. 1b. This figure complements Fig. 3b by considering 21 accessions without *C. ryukyuensis* ancestry. Note that the apomixis locus is located near the end of chromosome 1 (based on the Clementine reference sequence) which exhibits extensive MS admixture in common mandarins relative to other chromosomes. MS admixture is widespread in all sequenced mandarins. Two wild mandarins (M01=Daoxian wild mandarin and clonal relatives, M04=Suanpangan) show hybrid ancestry with nearly equal contribution from MS and MA. Source data underlying Fig. 4b and c are provided as a Source Data file.

We have shown that shiikuwasha, tachibana, and yukunibu are distinct homoploid (i.e., non-polyploid) hybrid species. Homoploid hybrid species are rare and are generally complex mixtures of parental species due to the absence of reproductive isolation between F1 hybrids and their parents[60,61]. Such introgressive hybridization promotes gene flow and the potential for fixing novel genetic combinations from the parental species. In contrast, shiikuwasha, tachibana, and yukunibu are fixed F1 hybrids that carry substantial genetic variation from their sexual parental species (*C. ryukyuensis*) on one haplotype, but only limited variation from their apomictic mainland parent(s) on the other. These hybrid genetic configurations are stably propagated by ongoing apomictic reproduction, which also reproductively isolates the hybrids from their parental species. This scenario provides a new model for homoploid hybrid speciation involving apomixis that could also apply to other plant taxa.

Our new conceptual framework for mandarin evolution and domestication illuminates the origin of other key traits besides apomixis. For example, loss of anthocyanin production is widespread among domesticated mandarins and some of their hybrid relatives[50], and results in the appealing white flowers celebrated in ancient poetry compared with the ancestral purple color seen in other citrus. While elegant genetic analyses have shown that this recessive trait is due to defective alleles of the MYB transcription factor *Ruby*[49,50], its evolutionary origin is unknown. Since the same deletion allele is fixed in both mangshanyeju and *C. ryukyuensis*, it was likely already fixed in the common Pleistocene ancestor of all mandarins. (An overlapping second deletion allele is now also segregating in common mandarins). It follows that loss of anthocyanin in mandarins preceded domestication (Supplementary Note 10). While adaptive introgressions from wild populations have played a notable role in crop and animal

domestication[62,63], the introgression of apomixis and anthocyanin loss from wild populations occurred prior to domestication of mandarins, highlighting the importance of wild alleles at all stages of domestication.

From an agronomic point of view, apomixis contributed to the spread of appealing phenotypes, accelerating domestication and impacting subsequent breeding strategies. This behavior is especially desirable in citrus and other woody plants with longer reproductive cycles. In our scenario, the domestication of mandarin citrus is characterized by three main events: (1) the rise of apomixis in the mainland mandarin lineage during Pleistocene, (2) pummelo introgression that incorporated desirable traits into the ancestral mandarin genome[21], and (3) the myriad of crosses between those ancestral hybrids and admixtures, that gave rise to the current basic types of edible citrus[20,58] (Supplementary Note 12). These findings provide inspiration for the breeding of new hybrids with disease-tolerance—for example, one of the top performing rootstocks tolerant to Huanglongbing (citrus greening disease) is a hybrid with shiikuwasha parentage[64]—as well as other desirable horticultural and nutritional characteristics[6,65].

## Methods

**Sampling of diverse east Asian citrus**. Currently available genomic data for mandarin (*C. reticulata*) and related citrus are concentrated on (1) cultivated varieties[17,20,21] and (2) wild varieties from southeastern China, the center of diversity for mandarin citrus[4]. The extensive wild and cultivated diversity in the Ryukyu islands and southern mainland Japan has been largely neglected with only two tachibana accessions included in previous collections[4,20].

To sample diverse east Asian citrus, we collected (1) named varieties of shiikuwasha and other traditional cultivated varieties (oto, kabuchi, tarogayo) from local Okinawan growers and the Okinawa Prefectural Agricultural Research Center, (2) unnamed local mandarin-type citrus trees from personal gardens and small farms, including the Katsuyama Shiikuwasha Co., Ltd., (3) wild citrus trees from various sources including naturally forested areas, including samples later identified as belonging to the new species *C. ryukyuensis*, (4) local island-grown trees of known or uncertain provenance from Okinawa World Theme Park, and (5) wild and cultivated tachibana from southern mainland Japan. Our collection also includes two distant relatives of citrus in the Rutaceae family (*Murraya paniculata* and *Toddalia asiatica*) that are not used in the present analysis. In total, 71 samples including 69 citrus were collected with appropriate permissions. More detailed information about these collections is provided in Supplementary Note 2 and Supplementary Data 2.

**Genome sequencing and genotyping**. Shoots or young leaves were collected from each sample in the field. Leaves were frozen with liquid nitrogen, and then crushed in a mortar. Total DNA was extracted from the frozen powder of leaves using a DNeasy® Plant Mini Kit (QIAGEN Co., Hilden, Germany). After libraries had been prepared with a KAPA HyperPlus Library Preparation Kit (F. Hoffmann-La Roche, Ltd., Basel, Switzerland) and NEBNext® Ultra™ II FS DNA Library Prep Kit for Illumina (NEW ENGLAND BioLabs, Inc., Ipswich, MA), 150 and 250 bp paired-end reads were obtained using a HiSeq 2500 Rapid v2, HiSeq 4000 and NovaSeq 6000 SP (Illumina, Inc., San Diego, CA). Each sample was sequenced at more than 30-fold redundancy.

Illumina paired-end reads from each accession (both new and from refs. [4,17,20,21]) were mapped to the haploid Clementine reference sequence v1.0[21] using BWA-MEM (version 0.7.8-r455)[66]. We used the Clementine because it is the best quality available mandarin-type reference genome, and therefore suitable for exploring variation in mandarin. It has previously been established[20,21] that Illumina data from diverse citrus species can be readily aligned to this reference sequence, and the average mapping rate across samples sequenced in this study is 97.4%. Duplicate reads were removed using picard MarkDuplicates (version 2.13.2). Variants were called using GATK HaplotypeCaller (version 3.7-0-gcfedb67)[67] with filtering based on read map quality, base quality, read depth and allele balance (Supplementary Note 2).

**Identification of ancestral populations**. In order to identify natural or pure species or sub-species, we sought groups of individuals whose genomes are as closely related to each other as currently recognized species and conversely lack high heterozygosity characteristics of interspecific hybridization. We used three complementary approaches, sliding window analysis of pairwise genomic distance, multidimensional scaling, and admixture analysis.

The distance measure $D$ between two diploid genomes 1 and 2 is defined by[21]

$$D = 1 - \frac{\pi_1 + \pi_2}{4\pi_{12}} \qquad (1)$$

where $\pi_1$ and $\pi_2$ are the respective heterozygosities (i.e., nucleotide diversity) of the two accessions, and $\pi_{12}$ is their sequence divergence (i.e., probability that randomly chosen alleles from the two diploids are different). The value of $D$ ranges from 0 to 1, with $D = 0$ for clones, $D = 0.5$ for two unrelated individuals from a panmictic population, and $D$ approaching 1 for two deeply divergent species.

We used a sliding window of 200 kb to calculate the distance D between two differentiated populations (e.g., different species or sub-species) taking one accession from each population, along with the heterozygosity of each genome. Genome wide values of $D$ consistently above 0.5 without abrupt changes in heterozygosity indicate pure genomes without admixture. Deviations from this pattern at certain windows suggest admixture for those genomic regions[21].

We performed multidimensional scaling analysis using the R (version 3.5.1) function cmdscale based on pairwise genomic distances ($D$ defined above). We first identified clones ($D \approx 0$) and chose the accession with the highest sequencing depth to represent each clonal group. Results for east Asian mandarin-type citrus are shown in Fig. 1a. Pure taxa (PU, RK, MS, MA) are found as corner clusters, and accessions lying between corners or near the middle of the diagram are found to be hybrids formed from these pure taxa.

We also identified ancestral populations and estimated genome-wide ancestry proportions using ADMIXTURE[68]. For mainland Asian citrus, the lowest cross-validation error was obtained with $K = 6$ corresponding to common mandarin (MA), mangshanyeju (MS), pummelo (PU), citron, *Fortunella* (kumquat), and Ichang papeda (Supplementary Fig. 4a, Supplementary Note 9). Since citron, *Fortunella*, and Ichang papeda are absent from our mandarin core set, they do not appear in the multidimensional scaling of Fig. 1a. ADMIXTURE analysis of the mandarin core set of Fig. 1a (i.e, omitting known citron, kumquat, papeda and their hybrids) also identifies *C. ryukyuensis*, common mandarin, and mangshanyeju to be distinct populations [data not shown]. Further evidence for the distinctness of mangshanyeju and common mandarin is found in clustering by pairwise genomic distance (Supplementary Fig. 4b). We note that accessions MS1 and MS2 are pure mangshanyeju, and M01 and M04 are F1 hybrids of mangshanyeju and common mandarin.

**Phylogenetic inference**. For phylogenetic inference of Asian citrus species based on nuclear genomes, we used single nucleotide polymorphisms in the introns and UTRs to minimize selection pressure bias. Each diploid genome was reduced to a haploid sequence by randomly sampling one allele at each variant position (the species phylogeny is insensitive to this sampling procedure). We required complete coverage across all representative individuals (i.e., no missing genotype calls) and recovered a total of 209,124 single nucleotide variable sites. Using these characters we constructed a maximum likelihood tree with RAxML[69] under the general time-reversal model of nucleotide substitution with 1000 bootstrap replicates ("raxmlHPC -m GTRGAMMA -N 1000"). The tree was rooted with Chinese box orange (*Severinia buxifolia*, also known as *Atalantia buxifolia*). The highly supported topology of this species tree (Supplementary Fig. 2a) is in agreement with our previously published nuclear genome phylogeny based on SNPs in complementary regions of the genome (non-genic, non-repetitive, and non-pericentromeric)[20], affirming the robustness of the tree topology.

We determined the chloroplast genotype of each accession by mapping reads to the chloroplast genome sequence of sweet orange[70]. This is an appropriate reference for genome wide pan-citrus comparisons. We constructed maximum likelihood phylogenetic trees using RAxML[69] under the general time-reversal model of nucleotide substitution with 1000 bootstrap replicates. (Supplementary Fig. 2b, Supplementary Note 2).

***C. ryukyuensis* allele frequency spectrum**. The allele frequency spectrum (AFS) for the Okinawa *C. ryukyuensis* population was computed based on single nucleotide polymorphisms of the eight distinct sequenced accessions, excluding short admixed genomic segments and using only sites with no missing data (Supplementary Fig. 3). For demographic inference, we used moments[71] to model the folded AFS to detect the possible existence of a population bottleneck. A likelihood ratio test between a panmictic constant effective population size model (no bottleneck) and a two-epoch model shows that the two-epoch model with a population bottleneck provides a better fit. Though the strength of the bottleneck cannot be determined based on the AFS alone due to the small sample size (Supplementary Note 3), it can be estimated in the context of a 4-population divergence model (Supplementary Note 11).

**Divergence time and effective population size estimates**. To estimate the population divergence times and effective population sizes of pummelos, *C. ryukyuensis*, mangshanyeju and common mandarins (Supplementary Fig. 5a), we first derived the joint allele frequency spectrum of eight accessions (two per population) from genomic regions without admixture. We implemented demographic inference using moments[71], a python package that can efficiently simulate multidimensional allele frequency spectrum and infer demographic history. Time calibration was based on a late Miocene citrus leaf fossil[72] and a previous estimate for mandarin-pummelo divergence[20] (Supplementary Note 11). For effective population size estimates, we used a generation time of 10 years. For moments simulations, multiple runs were performed with independent starting points in the high-

dimensional parameter space and checked for convergence of the likelihood and model parameter values. The estimated model parameters are listed in Supplementary Fig. 5b with uncertainties reflecting the time calibration range.

**Ancestry informative markers and local ancestry analysis.** In order to characterize the ancestry of mandarin types in detail, we first identified a genome-wide set of ancestry informative markers (AIMs) for four ancestral populations (RK=*C. ryukyuensis*, MS=mangshanyeju, MA=common mandarin, PU=pummelo) using pure, or mostly pure, individuals that were identified by sliding window analysis of pairwise genomic distance (D) and multidimensional scaling. AIMs were derived using three pure pummelos, three pure *C. ryukyuensis* accessions, two pure mangshanyeju, and three common mandarins as exemplars for the four ancestral populations. We note that our dataset only includes two pure mangshanyeju. We also found that common mandarins generally contained sub-chromosomal genomic segments with MS or PU ancestry. Segments of these exemplars with evidence for admixture were excluded. (Supplementary Note 3).

AIMs for each target population were defined as single nucleotide variants that are fixed in the target population exemplars relative to the other population exemplars, i.e., homozygous in the target exemplars but not found in the others. Since MS and MA are more weakly differentiated than other population pairs, we introduced a super-population MM to represent markers that are fixed in *C. reticulata* (combined MA and MS, together with mainland Asian mandarin) relative to PU and RK. In this way, we obtained a total of 397,887 ancestry informative markers: 268,383 for PU, 54,325 for RK, 35,067 for MA, 22,482 for MM, and 17,630 for MS.

We used these genome-wide AIMs to carry out local ancestry analysis in our collection of east Asian citrus (Figs. 1b, 3b, 4c). Sliding windows of 500 AIMs were employed and the ancestry for each window was assigned using a likelihood-based approach, following Wu et al.[20]. To call ancestry within a window, we required at least 5 AIMs for each ancestral population, otherwise the ancestry was assigned as Unknown. We note that our local ancestry method is in general agreement with but more sensitive than ADMIXTURE[68], which may fail to detect short blocks of admixture at the a few percent level.

**Haplotype sharing and familial relationships.** Genetic relatedness between a pair of diploid individuals can be quantified by the proportion of their genomes that share zero, one or two haplotypes that are 'identical by descent' (IBD0, IBD1, and IBD2). The familiar coefficient of relatedness is defined from these quantities via $r = \frac{1}{2}$ IBD1 + IBD2. Following[20,21], we infer identify by descent over non-overlapping 200 kb windows using (**1**) the genetic distance D defined above and (**2**) the identity-by-state ratio,

$$IBSR = IBS2/(IBS2 + IBS0) \qquad (2)$$

where IBS2 is the number of shared heterozygous sites in a window (i.e., joint-genotype AB|AB, sharing two different alleles identical-by-state), and IBS0 is the number of homozygous differences (i.e., joint-genotype AA|BB, no allele sharing). For individuals from the same population, IBSR is independent of allele frequencies and has a mean of 2/3 for two unrelated individuals if the population is panmictic[73]. Importantly, if two individuals share the same haplotype across a window, then IBS0=0 and IBSR=1. We infer the IBD state for each window using the following criteria[20]: If IBSR < 0.95, the genomic window is assigned IBD0. If IBSR >=0.95 and D < 0.05, the window is assigned IBD2. If IBSR >=0.95 and D > 0.05 the window is assigned IBD1.

Genomic windows for two interspecific hybrids need to be treated differently, since in these cases the IBSR value is inflated by species-specific alleles and does not reflect shared haplotypes[21]. For such regions, we inferred the IBD state by comparing phased haplotypes instead of diploid genotypes. We performed interspecific phasing using representative accessions from the two parental populations[20]. To allow errors from SNP calling and phasing, we consider two haplotypes identical if the mismatch rate is below $2 \times 10^{-4}$.

By this method we find that the mainland Asian mandarin RK3 shares one haplotype with all shiikuwasha across the entire genome. Since shiikuwasha are F1 interspecific hybrids, this implies that RK3 is the common parent of shiikuwasha (or more properly in light of its apomictic reproduction, a clone of the common parent). We found that the elite Nakamoto seedless shiikuwasha (SH4) is a somatic mutant of one of the six basic shiikuwasha genotypes (SH2) (Supplementary Note 4). Similarly, kunenbo-mikan shares one haplotype with oto, kabuchii, tarogayo, and two other accessions, implying that the kunenbo-mikan genotype is parental to these accessions (which we call the yukunibu group). The *C. ryukyuensis* haplotypes of various shiikuwashas and yukunibus are found to be unrelated by direct comparison.

For tachibana we could not identify a mainland mandarin parent in our collection, but interspecific phasing and haplotype sharing analysis shows that there are at most two distinct *C. reticulata* haplotypes across three distinct tachibana genotypes. This is consistent with a single mainland Asian mandarin ancestor. By contrast, each tachibana carries a distinct *C. ryukyuensis* haplotype (Supplementary Fig. 6a) implying different *C. ryukyuensis* parents. The pairwise genomic IBD proportions and the coefficient of relatedness are shown in Supplementary Fig. 6b (see also Supplementary Note 5).

Other familial relationships discovered in this work are described in Supplementary Notes 7 and 8.

**Genetic differentiation between extant and inferred *C. ryukyuensis* populations.** To measure the genetic differentiation between *C. ryukyuensis* populations, we estimated Weir-Cockerham's Fst with vcftools[74] using genomic regions without admixture. For the purpose of comparing progenitor *C. ryukyuensis* populations of shiikuwasha, tachibana, and yukunibu, we extracted *C. ryukyuensis* haplotypes from each group and formed pseudo-diploids before using vcftools to estimate Fst. As an alternative approach to calculate the genetic differentiation between the extant *C. ryukyuensis* population in Okinawa and the *C. ryukyuensis* ancestors of mainland Japan tachibana population, we compared diploid *C. ryukyuensis* segments of tachibana to the corresponding genomic regions in Okinawa *C. ryukyuensis* accessions (Supplementary Note 11).

**Characterizing apomixis haplotypes.** To investigate the genetic ancestry of the *CitRKD1* MITE-insertion allele and its connection to the mangshanyeju population, we examined the 200 kb region flanking the *CitRKD1* gene (Ciclev10010497m) (chromosome 1: 25,380,489–25,582,037 of the Clementine reference sequence[21]) across a collection of 55 citrus accessions including mandarins and interspecific mandarin hybrids and admixtures derived from six citrus species (PU=pummelo, CI=citron, IC=Ichang papeda, FO=Fortunella, RK=*C. ryukyuensis* and *C. reticulata*). We carried out genetic admixture analysis for this 200 kb window using ADMIXTURE[68] for K=3–11. For each value of K, we performed twenty independent runs and used the run with lowest cross-validation error. Figure 4b shows the population ancestry composition at K = 8, with additional plots for K = 8–10 included in Supplementary Fig. 8c (See Supplementary Note 10). For K = 8 all six citrus species are resolved with further differentiation of mandarins (*C. reticulata*) into three sub-populations, namely, common mandarin (MA), and two mangshanyeju sub-populations h1 and h2 across this window. Progressively finer resolution is observed for K = 9 and 10. At K = 9, two sub-populations (m1, m2) are differentiated within common mandarins. With ten ancestral populations (K = 10), further differentiation within pummelos (PU, p2) is revealed. Importantly, the two mangshanyeju (h1, h2) sub-population ancestry compositions remain unchanged for K = 8–10.

**Reporting summary.** Further information on research design is available in the Nature Research Reporting Summary linked to this article.

## Data availability

Data supporting the findings of this work are available within the paper and its Supplementary Information files. A reporting summary for this Article is available as a Supplementary Information file. High coverage (average 46x) whole-genome shotgun-sequencing data of 69 citrus accessions generated in this study have been deposited at NCBI under BioProject PRJNA670310, with summary information for each accession in Supplementary Data 1 and 2. Previously published resequencing data used in this study are listed in Supplementary Data 3. Source data are provided with this paper.

## Code availability

Custom wrapper scripts for demographic inference using moments are available at Github [https://github.com/citruscompgen/RyukyuCitrus.git].

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

## Acknowledgements

We are grateful to Hiroshi Kobashigawa for sharing his expertize and hospitality, where the first *C. ryukyuensis* tree was observed, and to Yoshikatsu Yamakawa and Hiromitsu Yasumura from Katsuyama Shiikuwasha Co., Ltd. for their assistance and providing the Katsuyama mandarin accession related to all shiikuwasha. We thank Shuji Takino and Hideki Yamashiro for assistance in sample collection. We thank Nate Jameson for contributing his citrus expertise in the field in Okinawa, and John Willis for helpful discussions. We thank staff from the Okinawa Institute of Science and Technology (OIST) DNA sequencing section, Onna, Okinawa for their help with the DNA-sequencing. This study was supported by OIST Internal Funds (D.S.R.). The collaborative efforts of G.A.W, D.S.R, and F.G.G are partially supported by a grant from the Citrus Research and Development Foundation (18-010), on behalf of the Florida citrus growers. The work conducted by the U.S. Department of Energy Joint Genome Institute, a DOE Office of Science User Facility, is supported under Contract No. DE-AC02-05CH11231 (G.A.W., D.S.R.). The work conducted at the Centro de Genomica at IVIA (Spain) is supported through grants No RTI2018-097790-R-100 (Ministerio de Ciencia, Innovación y Universidades/Fondo Europeo de Desarrollo Regional) and No 51915 (IVIA)(MT). D. S.R. is grateful for the support of the Marthella Foskett Brown Chair in Biological Sciences. We thank the DNA Sequencing Section, and the Information Service Section of OIST for technical support. We thank Shuji Takino for samples of TB0 and TB6 from Hashimoto and Shingu in Shingu City in Wakayama prefecture and for help during the sampling of TB3 and TB4 from Arima in Kumano city in Mie prefecture; Hirofumi Hamada for samples of TB7 from Suno in Kumano City in Mie prefecture; Naoe Ooe for samples of TB5, TB8 and YZN from Nikou in Nachikatsuura town in Wakayama prefecture; Motofumi Yamasaki for samples of YZS and DDS from Shingu in Shingu City in Wakayama prefecture; Hirofumi Higa for samples of SH0, SH1, SH2, SH3, OTO, KBU and TRG from Asahikawa in Nago City in Okinawa prefecture; Okinawa Prefectural Agricultural Research Center (OPARC) of Nago in Nago city in Okinawa prefecture for samples of SH4; Yoshikatsu Yamakawa from Katsuyama Shiikuwasha Co., Ltd. of Katsuyama in Nago City in Okinawa prefecture for samples of RK3, RK4, RK5, SH5 and SH6; Eijyun Zamami for samples of R15 and ISH from Izumi in Motobu Town in Okinawa prefecture; Masakazu Nakazato for samples of R02 from Katsuyama in Nago City in Okinawa prefecture; Hiroshi Kobashigawa for samples of SHI, RK1, RK2, RK7, RK8, RK9, TB1, TB2 and KB2 from Oogimi in Oogimi Village in Okinawa prefecture; Sachiko Miyagi for samples of RK6 from Oku in Kunigami Village in Okinawa prefecture; Hideki Yamashiro from Katsuyama Community Center on Katsuyama in Nago City in Okinawa prefecture for samples of R01 and R19; Tetsu Yonamine for samples of R07 from Kaneshi in Nakijin Village in Okinawa prefecture; Katsurou Kinjyou from a theme park called 'Daisekirinzan,' managed by Nanto Co., Ltd. of Ginama in Kunigami Village in Okinawa prefecture for samples of R05, R06 and R12; Kaori Miyazato for samples of R13 from Genka in Nago City in Okinawa prefecture; Yukihiro Shimabukuro for samples of R14 from Imadomari in Nakijin Village in Okinawa prefecture; Hideaki Oshiro for samples of R16, R17 and R18 from Katsuyama in Nago City in Okinawa prefecture; Chogi Miyagi for samples of R03 from Hama in Kunigami Village in Okinawa prefecture; Kiyoshi Miyagi for samples of R04 from Nuha in Oogimi Village in Okinawa prefecture; Noboru Teruya for samples of R00 from Tokijin in Nakijin Village in Okinawa prefecture; Tetsuo Shimabukuro for samples of KB1 from Inamine in Nago City in Okinawa prefecture; Tomohiro Chinen from a theme park called 'Okinawa World,' managed by Nanto Co., Ltd. of Tamashiro in Nanjyo City in Okinawa prefecture for samples of KB3, CAR, CI0, CI1, CI2, OBN, TK0 and TK1.

## Author contributions

G.A.W., F.G.G. Jr, and D.S.R. conceived and led the project with C.S. and M.T. C.S. led and performed sampling, sequencing, and historical analysis. G.A.W. led and performed sequence analysis with input from D.S.R. and consultation with M.T. and F.G.G. Jr. C.S., H.K. and C.A. carried out historical and cultural background analysis. C.S., H.K., M.T. and F.G.G. Jr. contributed to the description of citrus accessions and discriminatory characteristics. M.T. and G.A.W. led the discussion of timing and dispersal of mandarins and consequences for citrus domestication in discussion with F.G.G .Jr. and D.S.R. C.S. and C.A. were responsible for logistics and communication with growers and sources. H.K. contributed samples. F.M. contributed seedless shiikuwasha sample. G.A.W., C.S., M.T., F.G.G. Jr., and D.S.R. wrote the manuscript with input from all authors.

## Competing interests

The authors declare no competing interests.
