## [Peer Review File · Nature Communications]

REVIEWERS' COMMENTS

Reviewer #1 (Remarks to the Author):

I wish to commend the authors on the laudable job done in revising the text and address my concerns. The new text is a big improvement over the previous text, and I only have a few very minor edits and comments I have included in a track-changes version attached.

Reviewer #2 (Remarks to the Author):

The paper "The nature of mandarin citrus: species, hybridization, and apomixis" represents an incredible study in term of used plant material and in-depth analysis, contributing to explain the origin and the role of apomixis in the citrus domestication, as well as the origin of a new Citrus species. Mandarins are able to produce a progeny true-to-type due to nucellar embryony, also named apomixis, consisting in the propagation by seeds. Up to now, even though this trait is dominant and the locus controlling this trait has been recently discovered (Wang et al., 2017; Shimada et al., 2018), the origin and the diffusion among mandarins and their admixture is still unclear. The present works masterfully elucidate a new picture of the origin of mandarin and the apomixis inheritance. The identification of the new species *C. ryukyuensis* (representing the fourth group of ancient mandarin in addition to tachibana, shiikuwasha, yukunibu), the genome sequencing of many wild and domesticated mandarins from mainland Japan, Asia and Ryukyu island and the inheritance of cpDNA, in a fascinating and complex intertwining of lineages, cooperate in the elucidation of the widespread of mandarins and the origin of apomixis of mandarins. The main results can be following resumed in the identification of *C. ryukyuensis* as a new species; the identification of the origin of the apomixis allele; the Paleogeology support to the evolution; the domestication of mandarins.

In the last 15 years, the study of the citrus genome had a tremendous impact within the International Citrus Community and the main papers (Xu et al., 2014, Wu et al., 2014, Wu et al., 2018; Wang et al., 2017; Wang et al., 2018) improved, each time, in added value and details previous knowledge, demonstrating the maximum complexity of the Citrus genus. The first classification of mandarins has been further improved with the present manuscript, thanks to the resequencing of a large collection of wild and domesticated mandarins. Moreover, it was also better demonstrated the role and the introgression of mangshanjeyu, contributing to classify finely the domestication of mandarins. This new update was possible considering the new identification of *C. ryukyuensis*, inducing to describe varieties with *C. ryukyuensis* ancestry and use subtypes to designate different hybrid/admixture groups. Moreover, the early diversification of mandarin lineages across all of southeast and east Asia, and the origin and spread of apomixis represent one of the most impressive added values to the study.

The choice of plant material (mainly represented by broad sampling and rich historical documentation) is wide and exhaustive to reach conclusions that authors advanced. All data are clearly presented and they are extensive. I understand and agree with the choice of using Illumina technology to make direct and equal comparisons with preexisting data for Chinese accessions, as well as to approve the fact Illumina platform represent the more suitable technology to characterize variations within and between species, clarifying both familial relationships and population-level ancestry.

Conceptually the conclusions discussed by authors are robust, valid and relevant. I agree with the new reorganization of the manuscript addressed to emphasize the origin and diffusion of apomixis. I also appreciate the new focus and discussion on anthocyanins, aimed to show how the evolutionary framework is consistent with the emergence of the white flower phenotype in mandarin-type citrus, and that the loss of anthocyanin production clearly preceded domestication.

Now I found clearer the identification and the use of samples used in all the study, as reported in the new tables and figures. About *Murraya paniculata* e *Sarukake-mikan* I continue to find out of the scope their use, even though the idea of authors is to make them available to the plant community. If they do not add nothing more to the present paper and they are not directly used to draw conclusion to the study, I do not find sense to report them. If so, I suggest to eliminate them from the present

paper.

References are appropriately reported.

All the paper, considering the high complexity of the study, is clear and well presented

Reviewer #3 (Remarks to the Author):

I would like to thank the authors for their detailed responses to the reviewer comments and for essentially rewriting the manuscript. I think the revised manuscript is much easier to read and the significance of the results is clear.

Referees' comments are highlighted in blue, our responses are interspersed in black

REVIEWERS' COMMENTS

Reviewer #1 (Remarks to the Author):

I wish to commend the authors on the laudable job done in revising the text and address my concerns. The new text is a big improvement over the previous text, and I only have a few very minor edits and comments I have included in a track-changes version attached.

Thanks. We have revised the manuscript to incorporate most of the suggestions and comments:

- 1) Page 4: Where to find this result?
Response: we have added citation to Figure 1b.
- 2) Page 4: Could be other explanations for reduced het. Perhaps tone this done? "suggested" or something along those lines
response: we have changed to 'suggested' as suggested!
- 3) Page 5: Delete the word "migrant" in "This implies that the apomixis trait was transmitted to these three hybrid species by their migrant mainland mandarin parents".

Response: We prefer to include the word "migrant". The direction of gene flow is clearly from mainland Asia to the Ryukyus. The hybridizations must have occurred in the Ryukyu/Japanese islands, since no admixture of *C. ryukyuensis* is found elsewhere. Since the directionality of gene flow is clear, we prefer to indicate it here.

- 4) Page 7: This doesn't seem a particularly novel paradigm. I'd be tempted to remove this sentence.

Response: we have kept this sentence but simplified it. We introduced this approach with citrus in the works cited, and extend it here, so it seems worth pointing out.

- 5) Page 7: Paradigm? Maybe "avenue"? Or "perspective"? And maybe another half sentence saying why this matters? That people should consider apomixes a potential driver of homoploid hybrid speciation in other taxa

Response: we have changed the sentence from "This scenario provides a new paradigm for homoploid hybrid speciation" to "This scenario provides a new model for homoploid hybrid speciation involving apomixis that could also apply to other plant taxa"

Reviewer #2 (Remarks to the Author):

The paper “The nature of mandarin citrus: species, hybridization, and apomixis” represents an incredible study in term of used plant material and in-depth analysis, contributing to explain the origin and the role of apomixis in the citrus domestication, as well as the origin of a new Citrus species. Mandarins are able to produce a progeny true-to-type due to nucellar embryony, also named apomixis, consisting in the propagation by seeds. Up to now, even though this trait is dominant and the locus controlling this trait has been recently discovered (Wang et al., 2017; Shimada et al., 2018), the origin and the diffusion among mandarins and their admixture is still unclear. The present works masterfully elucidate a new picture of the origin of mandarin and the apomixis inheritance. The identification of the new species *C. ryukyuensis* (representing the fourth group of ancient mandarin in addition to tachibana, shiikuwasha, yukunibu), the genome sequencing of many wild and domesticated mandarins from mainland Japan, Asia and Ryukyu island and the inheritance of cpDNA, in a fascinating and complex intertwining of lineages, cooperate in the elucidation of the widespread of mandarins and the origin of apomixis of mandarins. The main results can be following resumed in the identification of *C. ryukyuensis* as a new species; the identification of the origin of the apomixis allele; the Paleogeology support to the evolution; the domestication of mandarins.

In the last 15 years, the study of the citrus genome had a tremendous impact within the International Citrus Community and the main papers (Xu et al., 2014, Wu et al., 2014, Wu et al., 2018; Wang et al., 2017; Wang et al., 2018) improved, each time, in added value and details previous knowledge, demonstrating the maximum complexity of the Citrus genus. The first classification of mandarins has been further improved with the present manuscript, thanks to the resequencing of a large collection of wild and domesticated mandarins. Moreover, it was also better demonstrated the role and the introgression of mangshanjeyu, contributing to classify finely the domestication of mandarins. This new update was possible considering the new identification of *C. ryukyuensis*, inducing to describe varieties with *C. ryukyuensis* ancestry and use subtypes to designate different hybrid/admixture groups. Moreover, the early diversification of mandarin lineages across all of southeast and east Asia, and the origin and spread of apomixis represent one of the most impressive added values to the study.

The choice of plant material (mainly represented by broad sampling and rich historical documentation) is wide and exhaustive to reach conclusions that authors advanced. All data are clearly presented and they are extensive. I understand and agree with the choice of using Illumina technology to make direct and equal comparisons with preexisting data for Chinese accessions, as well as to approve the fact Illumina platform represent the more suitable technology to characterize variations within and between species, clarifying both familial relationships and population-level ancestry.

Conceptually the conclusions discussed by authors are robust, valid and relevant. I agree with the new reorganization of the manuscript addressed to emphasize the origin

and diffusion of apomixis. I also appreciate the new focus and discussion on anthocyanins, aimed to show how the evolutionary framework is consistent with the emergence of the white flower phenotype in mandarin-type citrus, and that the loss of anthocyanin production clearly preceded domestication.

Now I found clearer the identification and the use of samples used in all the study, as reported in the new tables and figures. About *Murraya paniculata* e Sarukake-mikan I continue to find out of the scope their use, even though the idea of authors is to make them available to the plant community. If they do not add nothing more to the present paper and they are not directly used to draw conclusion to the study, I do not find sense to report them. If so, I suggest to eliminate them from the present paper.

References are appropriately reported.

All the paper, considering the high complexity of the study, is clear and well presented

Response: Thanks. Regarding if we should remove the two samples that are distant relatives of citrus and not used in this study, we added the following comment to the 2nd paragraph in “Methods”, highlighting the sentence “Our collection also includes two distant relatives of citrus in the Rutaceae family (*Murraya paniculata* and *Toddalia asiatica*) that are not used in the present analysis.”:

Dear Editor: is there any reason to delete this sentence? It seems harmless to deposit this data into the public record. If you prefer, this sentence can be deleted, and the next sentence altered to, “In total, 69 citrus samples were collected with appropriate permissions.”

Reviewer #3 (Remarks to the Author):

I would like to thank the authors for their detailed responses to the reviewer comments and for essentially rewriting the manuscript. I think the revised manuscript is much easier to read and the significance of the results is clear.

Response: Thanks.